# Design and Isokinetic Training Control Method of Leg Press Training Device

**Hongfei Yu [1], Hongbo Wang [1,2,3,*], Yaxin Du [1], Xinyu Hu [2] and Ziyu Liao [1]**

[1] Parallel Robot and Mechatronic System Laboratory of Hebei Province, Yanshan University, Qinhuangdao 066000, China

[2] Key Laboratory of Advanced Forging & Stamping Technology and Science of Ministry of Education, Yanshan University, Qinhuangdao 066000, China

[3] Academy for Engineering & Technology, Fudan University, Shanghai 200433, China

[*] Correspondence: hongbo_w@ysu.edu.cn

**Abstract:** Lower-limb function in elderly people gradually degenerates with age, and poor rehabilitation conditions preventing the elderly from receiving scientific rehabilitation training result in the decline of social labor force and the increased economic burden of the elderly. Aiming at the characteristics of the single function and the complex structure of an existing telescopic leg trainer combined with the needs of the application group, a new type of leg-stretching training device with multiple training modes for lower extremity extension and flexion of the elderly is proposed. A new mechanical structure and electrical system is designed. At the same time, the anti-resistance training man–machine model is analyzed, aiming at the isokinetic resistance training mode, and a training controller strategy based on a fuzzy synovial algorithm is proposed. Finally, the feasibility of the designed controller strategy and the proposed leg training device are verified by prototype experiments, which will guide further research.

**Keywords:** leg-press device; isokinetic training; robust control; current tracking

## 1. Introduction

With the increasing aging of society, functional degeneration of the hip, the knee, the ankle, and the lower-limb muscles directly affects the health and the quality of life of patients [1,2]. Degradation of lower-limb motor function in the elderly results in reduction of the scope of their activities, such as eating, work, social contact, and communication. Moreover, less rehabilitation training for the elderly, the inherent prejudice of common diseases in the elderly, and the lack of clear understanding of specific training content can easily lead to resistance to rehabilitation training. At present, there is no product on the market that is designed for the elderly's lower-extremity muscle strength, rehabilitation methods, and physical and mental guidance.

In rehabilitation treatment of knee-joint motor function, continuous passive motion (CPM) rehabilitation training has obvious effects on muscle groups involving knee flexion and extension [3–5]. Active resistance motion (ARM) training, as a medium and high-intensity training mode, has proved to be beneficial to patients' health [6,7]. Therefore, a leg-stretching training device that can help patients achieve CPM rehabilitation training and ARM training is one of the main means of realizing quick recovery [8–10].

There are three types of products in the field of thigh rehabilitation and fitness in the market. The first is the traditional CPM machine, such as the Fisiotek 2000 of Italy [11], the CPM of the lower-limb continuous passive training instrument of the United States [12], and the K2000 of ZEPU [13]. It focuses on CPM training and is a relatively simple and single-function portable piece of rehabilitation

equipment. The second is the common weight-balanced kicking training device, such as the ELEMENT series [14] and the Pure Strength kicking training device of Technogym in Italy [15,16]. It is only used for high-intensity constant impedance resistance training, which is heavy, expensive, and not suitable for the elderly or rehabilitative patients [17,18]. The third is a pneumatic rehabilitation training machine, such as a Finnish firm's product, which is used in various forms of resistance training but is expensive and lacks power, thus it is unable to adequately provide active and passive training [19–21].

For the defects of these products, a new, multi-functional leg-stretching training device is designed in this article, which can realize various training modes, such as active and passive training and resistance training in the form of equal speed or resistance. At the same time, a new drive scheme combining motor and magnetic powder clutch is proposed. In addition, the related dynamics analysis and the design of the related anti-resistance synovium control algorithm are carried out. The feasibility of the control algorithm is verified by the experimental results. Reasonable resistance training can be carried out according to the characteristics of the limbs of the elderly [22].

## 2. Design of the Leg Press Training Device

### 2.1. Mechanical Structure Design

The leg-stretching training device designed in this paper is driven by a combination of motor and magnetic powder clutch, and the transmission scheme principle is shown in Figure 1. Compared with the traditional single-motor torque drive, the magnetic powder clutch is automatically clutched by the force difference between the two ends without additional signal control. The transmission torque can be easily controlled, which is very suitable for lower-limb rehabilitation robots [23]. In addition, the magnetic powder clutch has a fast response speed, low noise, controllable return speed, and high stability.

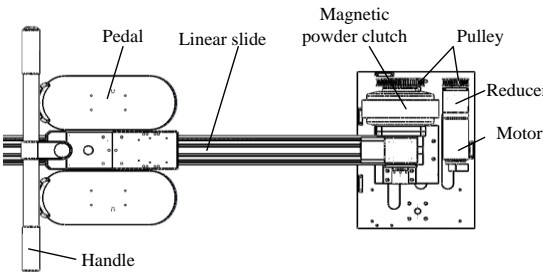

**Figure 1.** Structure of drive system.

At the same time, the motor and the clutch each perform their own functions, thus safety is higher. In addition, through different combinations of motor speed and transmission torque of the magnetic powder clutch, multiple training modes of a single piece of equipment can be realized. The use efficiency and range of the training device are improved, and the training modes corresponding to their combination modes are shown in Table 1.

**Table 1.** Control method of different training modes.

| Training Mode | Motor | Magnetic Powder Clutch |
|---|---|---|
| Passive training | Periodic positive reversal | Always set the highest torque |
| Active training | Change direction and speed of movement with pedaling force | Always put safe torque |
| Constant impedance training | Always reverse | Manually set constant torque |
| Constant velocity resistance training | Always reverse | Adaptive adjustment torque |

The leg-training device mainly consists of three parts: the patient part, the training part, and the power part, as shown in Figure 2. To suit the needs of different patients, the backrest elevation and the height of the patient's seat are adjustable. The training part is composed of a linear sliding rail, a push-pull component of the upper limb, and a pedal component. To facilitate the separation and the transportation of the seat and the training device body (including the training part and the power part), the supporting wheel and the connecting frame that is provided with a seat are set up. The power part is composed of a power box, which provides a corresponding driving force in different training modes. Meanwhile, the training speed of the pedals in the training area can be adjusted in real time through the encoder so as to ensure the safety and scientificity of the training process.

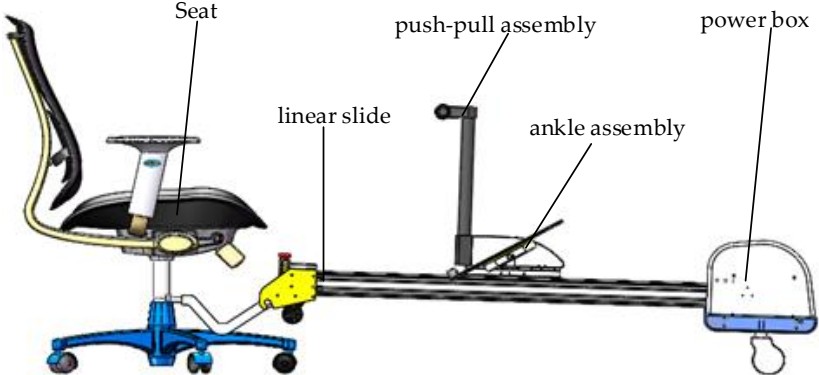

**Figure 2.** Mechanical structure of leg-press device.

## 2.2. Electrical System Design

The electrical system design of the leg-press device is shown in Figures 3 and 4. The main control system is based on the Stm32 microcomputer, and the communication system is based on Controller Area Network.

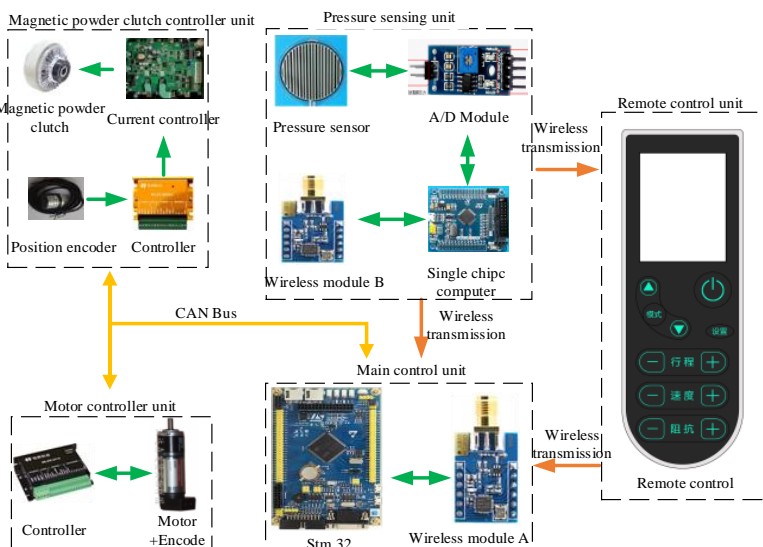

**Figure 3.** Electrical system.

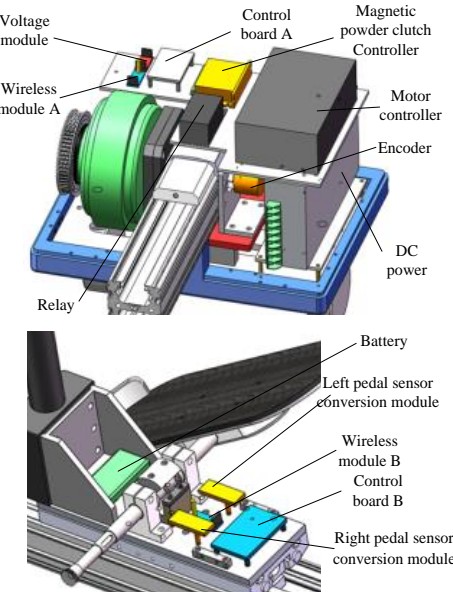

**Figure 4.** Controller distribution.

## 3. Human Dynamics Model of Resistance Training

### 3.1. Dynamic Model Solution of Mechanical Part

When the clutch slips and the torque does not exceed the maximum transfer torque, the transfer torque of the magnetic powder clutch can be seen from its static characteristics:

$$T_c = K_c(I_c - I_{c0}) \tag{1}$$

where $K_c$ means torque constant of the magnetic particle clutch, $I_c$ means the current of the magnetic powder clutch, and $I_{c0}$ means the starting current of the magnetic powder clutch.

When the magnetic powder clutch is in the engagement state, it is equal to the load torque.

$$T_c = T_h - \text{sgn}(v_p)T_f + J_{co}\frac{dw_{co}}{dt} \tag{2}$$

where $T_h$ means the torque of human pedal force acting on output axis of clutch, $T_f$ means the torque of slideway friction acting on clutch output shaft, $v_p$ means the pedal speed, $J_{co}$ means the rotating inertia of output shaft of clutch, and $w_{co}$ means the angular speed of output shaft of clutch.

The general dynamic equation of the mechanical system is obtained.

$$\begin{cases} \eta_1\eta_2 T_m - K_c(I_c - I_{c0}) = J_m\frac{dw_m}{dt} + J_d\frac{dw_d}{dt} + J_{ci}\frac{dw_{ci}}{dt}, lock = 0 \\ \eta_1\eta_2 T_m - \frac{1}{\eta_3}\left(T_p - \frac{v_p}{|v_p|}T_f\right) = J_m\frac{dw_m}{dt} + J_d\frac{dw_d}{dt} + J_{ci}\frac{dw_{ci}}{dt} + J_{co}\frac{dw_{co}}{dt}, lock = 1 \end{cases} \tag{3}$$

where *lock* means clutch sliding wear and clutch engagement, $\eta_1$, $\eta_2$ and $\eta_3$ mean the transmission efficiency of the reducer, synchronous pulley, and linear sliding rail, respectively, $T_m$ means the driving torque of the motor, $I_c$ means the exciting current of the clutch, $J_m$ and $w_m$ mean the rotational inertia and the angular velocity of the motor output shaft, respectively, $J_{ci}$ and $w_{ci}$ mean the rotational inertia and the angular velocity of the reducer output shaft, respectively, $J_{co}$ and $w_{co}$ mean the rotational inertia and the angular velocity of the clutch input shaft, respectively, $T_p$ and $T_f$ mean angular velocity of the output shaft of the clutch and the torque of the pedal force and the sliding friction force acting on the output shaft of the clutch, respectively, and $v_p$ indicates the pedal speed.

### 3.2. Dynamic Model of Human Leg Connecting Rod in Training Process

The human legs (thighs, legs, and feet) are regarded as rigid link. The foot on the pedal is always fixed in the same position (fixed pedal angle). Thus, the integral mechanism of the "human leg-slider" can be regarded as a crank-connecting rod mechanism. As shown in Figure 5, the human leg is simplified to a three-link rigid mechanism.

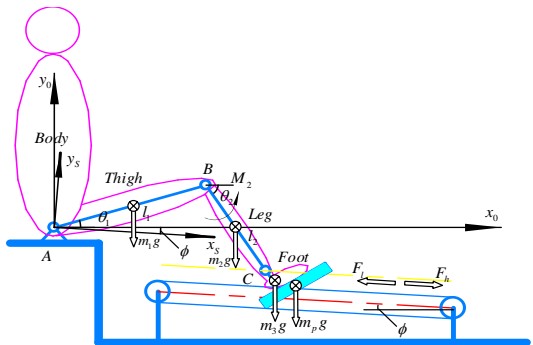

**Figure 5.** Human leg model during impedance training.

In Figure 5, $m_1$, $m_2$, $m_3$, and $m_p$ mean the mass of thigh, leg, foot, and slider, respectively, $F_l$ is the impedance force, and $F_h$ represents the pedal thrust along the slide.

Based on the principle of virtual work, the dynamic mathematical model of the mechanism is established. The mechanism is a system of mass points with complete constraints, which consists of three components' mass centers and the end point of the human leg under external force. Before listing the virtual work equation, the centroid parameters of the system of mass point of a man–machine system are first described, as shown in Tables 2–4.

**Table 2.** Centroid location description of components.

| Link Number | X-Coordinate $R_x^i$ | Y-Coordinate $R_y^i$ |
| --- | --- | --- |
| $l_1$ | $\frac{1}{2}l_1 \cos\theta_1$ | $\frac{1}{2}l_1 \sin\theta_1$ |
| $l_2$ | $S\cos\phi - e\sin\phi - \frac{1}{2}l_2\cos\theta_2$ | $-S\sin\phi - e\cos\phi - \frac{1}{2}l_2\sin\theta_2$ |
| $l_3$ | $S\cos\phi - e\sin\phi + \frac{1}{2}l_3\cos(\theta_3+\phi)$ | $-S\sin\phi - e\cos\phi + \frac{1}{2}l_3\sin(\theta_3+\phi)$ |

**Table 3.** Description of centroid velocity of components.

| Link Number | X-Directional Velocity $\frac{dR_x^i}{dt}$ | Y-Directional Velocity $\frac{dR_y^i}{dt}$ |
| --- | --- | --- |
| $l_1$ | $-\frac{1}{2}f_1 l_1 \dot{S}\sin\theta_1$ | $\frac{1}{2}f_1 l_1 \dot{S}\cos\theta_1$ |
| $l_2$ | $\left(\cos\phi + \frac{1}{2}f_2 l_2\sin\theta_2\right)\dot{S}$ | $-\left(\sin\phi + \frac{1}{2}f_2 l_2\cos\theta_2\right)\dot{S}$ |
| $l_3$ | $\dot{S}\cos\phi$ | $-\dot{S}\sin\phi$ |

**Table 4.** Description of centroid acceleration of components.

| Link Number | X-Directional Acceleration $\frac{\partial}{\partial t}\left(\frac{dR_x^i}{dt}\right)$ | Y-Directional Acceleration $\frac{\partial}{\partial t}\left(\frac{dR_y^i}{dt}\right)$ |
| --- | --- | --- |
| $l_1$ | $-\frac{1}{2}l_1(f_1\sin\theta_1\ddot{S} + (f_1^2\cos\theta_1 + g_1\sin\theta_1)\dot{S}^2)$ | $\frac{1}{2}l_1(f_1\cos\theta_1\ddot{S} + (-f_1^2\sin\theta_1 + g_1\cos\theta_1)\dot{S}^2)$ |
| $l_2$ | $(\cos\phi + \frac{1}{2}l_2 f_2\sin\theta_2)\ddot{S} + \frac{1}{2}l_2(f_2^2\cos\theta_2 + g_2\sin\theta_2)\dot{S}^2$ | $-(\sin\phi + \frac{1}{2}l_2 f_2\cos\theta_2)\ddot{S} + \frac{1}{2}l_2(f_2^2\sin\theta_2 - g_2\cos\theta_2)\dot{S}^2$ |
| $l_3$ | $\ddot{S}\cos\phi$ | $-\ddot{S}\sin\phi$ |

In Tables 2–4, $S = AC$, and the values of $f_1$, $f_2$, c$g_1$, $g_2$ are as follows:

$$f_1(\theta_1, \theta_2) = \frac{\sin(\theta_2 - \phi)}{l_1 \sin(\theta_2 - \theta_1)} \tag{4}$$

$$g_1(\theta_1, \theta_2) = \frac{l_2 \cos(\theta_1 + \theta_2)\cos^2(\theta_2 + \phi) + l_1 \cos(2\theta_2)\cos^2(\theta_1 + \phi)}{l_1^2 l_2 \cos\theta_2 \sin^3(\theta_2 - \theta_1)} \tag{5}$$

$$f_2(\theta_1, \theta_2) = \frac{\sin(\theta_1 - \phi)}{l_2 \sin(\theta_1 - \theta_2)} \tag{6}$$

$$g_2(\theta_1, \theta_2) = \frac{l_1 \cos(\theta_1 + \theta_2)\cos^2(\theta_1 + \phi) + l_1 \cos(2\theta_1)\cos^2(\theta_2 + \phi)}{l_1 l_2^2 \cos\theta_1 \sin^3(\theta_1 - \theta_2)} \tag{7}$$

According to the principle of virtual work, it can be obtained as follows:

$$\delta W_e = -m_1 g \delta R_y^1 - m_2 g \delta R_y^2 - (m_3 + m_p)g\delta R_y^3 + M_2\delta\theta_2 - F_l\delta S \tag{8}$$

where $m_1$, $m_2$, $m_3$, and $m_p$ mean the weight of thighs, legs, feet, and slider, respectively, $F_l$ means resistance force, $M_2$ is a function showing the relationship between $\theta_2$, $\frac{d\theta_2}{dt}$, and quadriceps stimulation rate $Z$, and its equation is expressed as:

$$M_2 = -KZ\theta_2 - BZ\frac{d\theta_2}{dt}, \tag{9}$$

where $KZ$ means the extension of knee-joint muscle elasticity, and $BZ$ means the knee-joint extension damping.

$$\begin{aligned}\delta W_i &= m_1 a_x^1 \delta R_x^1 + m_1 a_y^1 \delta R_y^1 + J_1\ddot{\theta}_1\delta\theta_1 + m_2 a_x^2\delta R_x^2 + m_2 a_y^2\delta R_y^2 \\ &+ J_2\ddot{\theta}_2\delta\theta_2 + (m_3 + m_p)\ddot{S}\delta S\end{aligned} \tag{10}$$

Through Equations (8) and (10) and Tables 2–4, the following equation can be obtained:

$$\begin{aligned}&-\tfrac{1}{2}m_1 g f_1 l_1 \cos\theta_1 + \tfrac{1}{2}m_2 g f_2 l_2 \cos\theta_2 + (m_2 + m_3 + m_p)g\sin\phi \\ &-F_l - (KZ\theta_2 + f_2 BZ\dot{S}) = \tfrac{1}{4}m_1 f_1 l_1^2 \cos\theta_1\left(f_1\ddot{S} + g_1\dot{S}^2\right) \\ &+ m_2\left(\cos\phi + \tfrac{1}{2}f_2 l_2 \sin\theta_2\right)\left(\cos\phi\ddot{S} + \tfrac{1}{2}l_2 f_2 \sin\theta_2\ddot{S} + \tfrac{1}{2}l_2 f_2^2\cos\theta_2\dot{S}^2 + \tfrac{1}{2}l_2 g_2\sin\theta_2\dot{S}^2\right) \\ &+ m_2\left(\sin\phi + \tfrac{1}{2}f_2 l_2 \cos\theta_2\right)\left(\sin\phi\ddot{S} + \tfrac{1}{2}l_2 f_2 \cos\theta_2\ddot{S} + \tfrac{1}{2}l_2 f_2^2\sin\theta_2\dot{S}^2 - \tfrac{1}{2}l_2 g_2\cos\theta_2\dot{S}^2\right) \\ &+ J_1 f_1\left(f_1\ddot{S} + g_1\dot{S}^2\right) + J_2 f_2\left(f_2\ddot{S} + g_2\dot{S}^2\right) + (m_3 + m_h)\ddot{S}\end{aligned} \tag{11}$$

### 3.3. Dynamic Model of Human Leg Link in Training Process

When the clutch is in a sliding state on a slide of 125 cm in length, the training force (i.e., resistance) provided by the human leg motion device can be calculated as follows:

$$F_l = \frac{2\pi K_c(I_c - I_{c0})}{125} + \text{sgn}(v_p)\frac{2\pi T_f}{125} + \frac{2\pi J_{co}}{125}\dot{S} \tag{12}$$

The mathematical model of resistance training system can be shown as follows:

$$A(\theta_1, \theta_2)\ddot{S} + B(\theta_1, \theta_2)\dot{S}^2 + C(\theta_1, \theta_2)\dot{S} = K_{ce}I_c + D(\theta_1, \theta_2, Z) \tag{13}$$

where:

$$\begin{aligned}A(\theta_1, \theta_2) &= \tfrac{1}{4}m_1 f_1^2 l_1^2 + m_2 f_2 l_2 \sin(\theta_2 + \phi) \\ &+ \tfrac{1}{4}m_2 f_2^2 l_2^2 + J_1 f_1^2 + J_2 f_2^2 + (m_2 + m_3 + m_h)\end{aligned} \tag{14}$$

$$\begin{aligned}B(\theta_1, \theta_2) &= \tfrac{1}{4}m_1 f_1 g_1 l_1^2 + \tfrac{1}{2}m_2 l_2\left(\cos\phi + \tfrac{1}{2}f_2 l_2 \sin\theta_2\right)\left(f_2^2\cos\theta_2 + g_2\sin\theta_2\right) \\ &+ \tfrac{1}{2}m_2 l_2\left(\sin\phi + \tfrac{1}{2}f_2 l_2 \cos\theta_2\right)\left(f_2^2\sin\theta_2 - g_2\cos\theta_2\right) + J_1 f_1 g_1 + J_2 f_2 g_2\end{aligned} \tag{15}$$

$$C(\theta_1, \theta_2) = f_2^2 BZ \sin \sigma + \frac{2\pi J_{co}}{125} \tag{16}$$

$$D(\theta_1, \theta_2, Z) = \frac{2\pi K_c I_{c0}}{125} - \text{sgn}(v_p)\frac{2\pi T_f}{125} - KZ\theta_2$$
$$-\tfrac{1}{2}m_1 g f_1 l_1 \cos \theta_1 + \tfrac{1}{2}m_2 g f_2 l_2 \cos \theta_2 + (m_2 + m_3 + m_h)g \sin \phi \tag{17}$$

In the above kinetic equation, the sign of $K_{ce}$ is negative, and the sign of $K_{ce}I_c$ is always negative. The sign of $D(\theta_1, \theta_2, Z)$ items is mainly related to the value of $Z$. In general training, the leg muscle is in a training state, represented by $D(\theta_1, \theta_2, Z(Train)) > |K_{ce}I_c|$, and the pedal speed is positive. When returning, the patient's leg muscle is at rest, represented by $D(\theta_1, \theta_2, Z(Rest)) < |K_{ce}I_c|$, and the pedal speed is negative.

## 4. Resistance Training Controller Algorithm

### 4.1. Process Control Analysis of Resistance Training

Figure 6 shows a complete resistance training process, where $v_p$ is the pedal speed, $v_d$ is the set training speed, and $v_m$ is the motor speed. Stage $A$ is the starting stage of the leg extension, with the pedal accelerating but remaining below the training speed, and the clutch is in the slipping state at this time. Stage $B$ is the constant speed leg extension stage, at which point the pedal reaches the training speed and maintains a constant speed, and the clutch is in the slip state at this time. Stage $C$ is the slow stop stage of leg extension with the pedal decelerating above zero, and the clutch state is unchanged. Stage $D$ is the stage of bending leg transition with the pedal accelerating inversely but below the motor speed, and the clutch is in an unsynchronized state. Stage $E$ is the stage of constant speed leg bending, in which the pedal speed is always the motor speed, and the clutch is in a synchronous state. Stage $F$ is the slow stop stage of bending legs with the pedal decelerating but remaining above zero, and the clutch is in an unsynchronized state at this time.

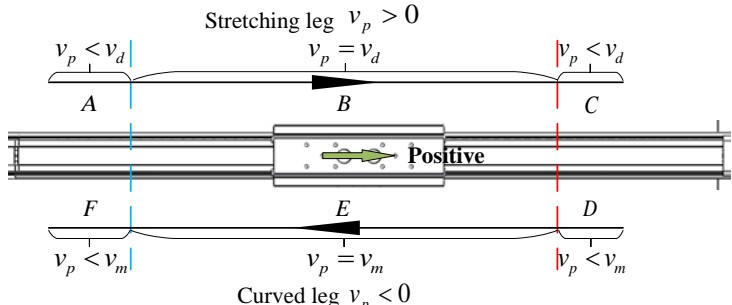

**Figure 6.** Stretching leg speed analysis.

In resistance training, isokinetic control only acts on the patient's muscle tension phase, and muscle tension or relaxation depend on the patient's consciousness. When the patient wants to finish a training return, if the control goal of the system is to maintain a constant pedal speed, the system is unstable. The reason is that there is no power for the pedal to advance, which is the patient's power. The system has no power source and cannot provide forward power. Therefore, it is necessary to distinguish the control objectives of each stage of the system. The stimulation rate $Z$ stimulates the joint torque to present two states: the training torque (assumed to be fixed) and the rest torque (assumed to be fixed). Table 5 shows the system motion parameters at different stages.

In the table, $f_{train}^*$, $f_{rest}^*$, $b^*$, and $m^*$ represent the pedal-to-slide force of training state, the pedal-to-slide force of rest state, the control term coefficient, and the system quality constant, respectively. According to the information in Table 5, it is specified that $\chi_1$ (consciousness is in the state of leg extension) is the training stage, and the system control objective function is $\dot{s} = v_d$. The $\chi_2$ segment (consciousness is in the curved leg) is the rest phase, and the system control objective function is $T_c = T_{cmin}$; $T_{cmin}$ is the return safe torque.

**Table 5.** Analysis of different training stages.

| Training Stage | State of Consciousness | Joint Torque $T$ | Pedal Speed $\dot{S}$ | Pedal Acceleration $\ddot{S}$ |
|:---:|:---:|:---:|:---:|:---:|
| A | Stretching leg | $T_{train}$ | $0 \rightarrow v_d$ | $\frac{f^*_{train} - b^* u}{m^*}$ |
| B | Stretching leg | $T_{train}$ | $v_d$ | $0$ |
| C | Curved leg | $T_{rest}$ | $v_d \rightarrow 0$ | $\frac{f^*_{rest} - b^* u}{m^*}$ |
| D | Curved leg | $T_{rest}$ | $0 \rightarrow v_m$ | $\frac{f^*_{rest} - b^* u}{m^*}$ |
| E | Curved leg | $T_{rest}$ | $v_m$ | $0$ |
| F | Stretching leg | $T_{train}$ | $v_m \rightarrow 0$ | $\frac{f^*_{train} - b^* u}{m^*}$ |

*4.2. Sliding Mode Variable Structure Controller Design*

The dynamic Equation (2) is shown as follows:

$$\ddot{S} = f_1 \dot{S}^2 + f_2 \dot{S} + GU + d \tag{18}$$

where $f_1 = -BA^{-1}$, $f_2 = -CA^{-1}$, $G = K_{ce}A^{-1}$, $d = DA^{-1}$, and $U = I_c$ are the control quantities. Because the actual man–machine system model is difficult to identify and some system model parameters are uncertain and time-varying, here, the certain and the uncertain parameters of the system are considered separately, and we can further obtain:

$$\begin{aligned}
\ddot{S} &= (f_{1n} + f_{1un})\dot{S}^2 + (f_{2n} + f_{2un})\dot{S} + (G_n + G_{un})U + d \\
&= f_{1n}\dot{S}^2 + f_{2n}\dot{S} + G_n U + \Delta
\end{aligned} \tag{19}$$

where $f_{1n}$, $f_{2n}$, and $G_n$ represent the certain parameters of the system, respectively, $f_{1un}$, $f_{2un}$, and $G_{un}$ represent the uncertain parameters of the system, respectively, $d$ is the system interference term, $\Delta$ represents the sum of the system uncertainties superposition. The $D(\theta_1, \theta_2, Z)$ was introduced in Equation (8), $K$, $B$, $X_0$, and $Z$ introduced in Equation (9) are related to the physical condition of the trainer itself. These parameters are unknown and different (each person is different), and Z means muscle stimulation rate and is characterized by time variance.

$$\Delta = f_{1un}\dot{S}^2 + f_{2un}\dot{S} + G_{un}U + d \tag{20}$$

Suppose that the sum of the uncertainties superposition is bounded:

$$|\Delta| \leq \delta, \delta \in R^+ \tag{21}$$

First, we can define the system tracking error $e = \dot{S} - \dot{S}_d$ and its derivative $\dot{e} = \ddot{S} - \ddot{S}_d$. Transferring them to Equation (19), we can obtain:

$$\dot{e} = -\ddot{S}_d + f_{1n}\left(e + \dot{S}_d\right)^2 + f_{2n}\left(e + \dot{S}_d\right) + G_n U + \Delta \tag{22}$$

Define $s = e$ as the sliding surface function, then $s = e$. For the linear and the nonlinear part of the system dynamics equation, the sliding mode control quantity is divided into two parts [24,25]; one is the equivalent control quantity of the first approximation system, and the other is the robust control quantity to deal with the nonlinear term [26,27].

$$U_s = U_{eq} + U_{hit} \tag{23}$$

The equivalent control quantity of the first approximation system is obtained.

$$\begin{aligned}
U_{eq} &= K_{ce}^{-1} A\left[\ddot{S}_d + BA^{-1}\left(e + \dot{S}_d\right)^2 + CA^{-1}\left(e + \dot{S}_d\right)\right] \\
&= K_{ce}^{-1}\left[A\ddot{S}_d + B\left(e + \dot{S}_d\right)^2 + C\left(e + \dot{S}_d\right)\right]
\end{aligned} \tag{24}$$

The nonlinear control quantity is obtained.

$$U_{hit} = -K_{ce}^{-1}A(\delta + \sigma)\text{sgn}(s) \tag{25}$$

where $\sigma$ is a real number greater than zero.

Therefore, the sliding mode control of resistance training can be rewritten as the following:

$$U_s = K_{ce}^{-1}\left[\begin{array}{c} A\left(\ddot{S}_d - (\delta + \sigma)\text{sgn}(e)\right) \\ +B\left(e + \dot{S}_d\right)^2 + C\left(e + \dot{S}_d\right) \end{array}\right] \tag{26}$$

In the equation, the switching function is composed of the system disturbance boundary $\delta$ and the robust coefficient $\sigma$. Both of these are fixed values. When there is a large error, the large control effect helps to reduce the error rapidly, but when the error is extremely small, the smallest control effect will destroy the stability of the system.

In order to eliminate the jitter caused by the robust term $\sigma\text{sgn}(s)$, this function term is slightly improved. That means setting the switching interface of the switching law-reaching law, $S = \pm r, r \in R^+$, through the segmentation control.

In the $S \geq \pm r$ region, the control function of the robust term $\sigma\text{sgn}(s)$ is maintained to ensure the fast convergence of the error. In the $S \leq \pm r$ region, a derivative term $\dot{s}$ is introduced to soften the control signal, weaken frequent overshoot, and reduce or avoid chattering. The improved function is as follows:

$$\text{sat}(s) = \left\{ \begin{array}{l} \text{sgn}(s), |s| > r \\ \frac{s}{r} + \beta\dot{s}, |s| \leq r \end{array} \right. \tag{27}$$

where $T\sigma \leq r \leq 1.1T\sigma$.

When $|s| \leq r$, the essence of the $\text{sat}(s)$ is used proportion and derivative (PD) control [28].

### 4.3. Fuzzy Identifier Design

The fuzzy controller in an integral form is used as the system disturbance identifier to control the uncertain term of the nonlinear part of the system. The structure of the fuzzy sliding mode resistance training controller is shown in Figure 7.

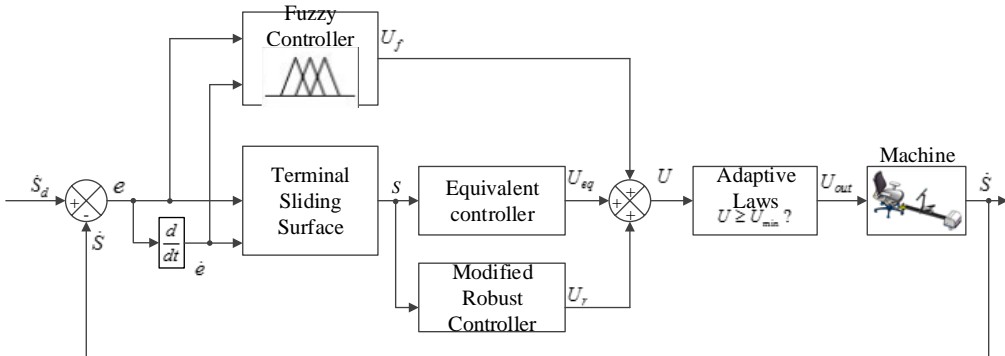

**Figure 7.** Structure of fuzzy sliding mode resistance training controller.

Compared with the resistance training controller of the sliding mode variable structure, the fuzzy sliding mode resistance training controller makes use of the fuzzy rule library, which replaces the switching function part of the output function of the sliding mode control. The switching function takes the system disturbance limit $\delta$ as the coefficient. When the system is on the sliding surface and is not affected by external disturbances, the equivalent control of the system will ensure that the system continues to move along the sliding surface S = 0. However, when the system deviates from the sliding

mode surface due to external interference or the influence of uncertain factors, the output of fuzzy control will drive the system to re-enter the sliding mode surface [29,30].

The disturbance compensation control quantity is represented by $U_f$. When people start training, there is a fixed minimum training force, which is $f^*_{cmin}$. Therefore, the initial value of the fuzzy control amount is set as $f^*_{cmin}$ and finally:

$$\begin{aligned} U_f(i+1) \quad &= U_f(i) + \Delta U_f(i+1) \\ &= f^*_{cmin} + \Delta U_f(1) + \cdots + \Delta U_f(i+1) \end{aligned} \tag{28}$$

where $i$ means the current sampling period.

System error $e$ and its derivative $\dot{e}$ are input as fuzzy controller, and $\Delta U_f$ is output as fuzzy control increases. The value of the error fuzzy domain is $[-3, 3]$, the unit of measurement is millimeters, and the error rate $\dot{e}$ represents the change rate of the leg speed. According to the design parameters of the device, the rate of change of velocity is $[-1, 1]$. The unit is millimeters per second, the basic domain of the $\Delta U_f$ is $[-10, 10]$, and the unit is mA. Take the area center of gravity method as a clear method. The fuzzy control rule table of $\Delta U_f$ is shown in Table 6.

**Table 6.** Fuzzy rule.

| E/DE | NB | NM | NS | ZO | PS | PM | PB |
|------|----|----|----|----|----|----|----|
| NB | PB | PB | PB | PM | PM | PS | ZO |
| NM | PB | PB | PB | PM | PS | ZO | NS |
| NS | PB | PB | PM | PS | ZO | NS | NM |
| ZO | PB | PM | PS | ZO | NS | NM | NB |
| PS | PM | PS | ZO | NS | NM | NB | NB |
| PM | PS | ZO | NS | NM | NB | NB | NB |
| PB | ZO | NS | NM | NM | NB | NB | NB |

Among them, NB, NM, NS, ZO, PB, PM and PS mean Negative Big, Negative Medium, Negative Small, Zero, Positive Big, Positive Medium, and Positive Small, respectively.

## 5. Resistance Training Control Experiment

In order to further verify the feasibility of the control algorithm, the experiment of isokinetic resistance training control was carried out on an experimental prototype. Figure 8 shows the situation of the isokinetic resistance training experiment conducted by different subjects. Data of the three subjects are shown in Table 7.

In Table 7, thigh length refers to the total length of the human lower limb from hip joint to knee joint, and leg length refers to the total length of the human lower limb from knee joint to foot bottom.

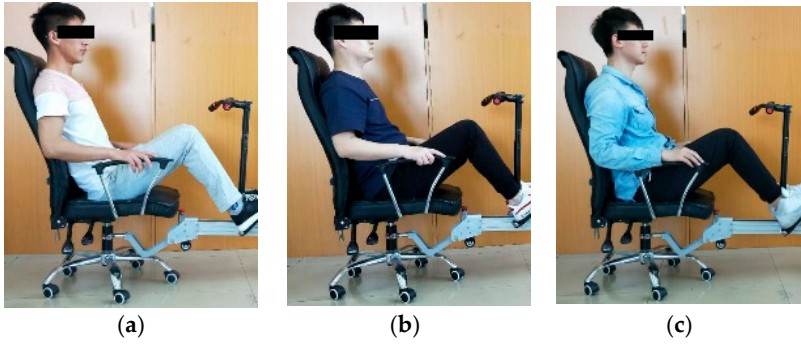

(a)　　　　　　　　　　　(b)　　　　　　　　　　　(c)

**Figure 8.** Constant velocity resistance training control experiment (**a**–**c**).

**Table 7.** Subject-related data.

| Subject $i$ | Weight (kg) | Height (cm) | Thigh Length (cm) | Calf Length (cm) | Pedaling Range (N) |
|---|---|---|---|---|---|
| A | 62 | 175 | 44 | 48 | 145~170 |
| B | 72 | 173 | 42 | 46 | 190~210 |
| C | 78 | 177 | 45 | 49 | 200~225 |

Three subjects were selected to conduct two groups of isokinetic resistance training (the target training speed was 160 mm/s and 200 mm/s, respectively). The experiment was repeated five times in each group. The sampling results of pedal speed are shown in Figure 9 (the sampling interval was 100 ms).

When the preset speed was 160 mm, subjects A, B, and C reached the preset training speed at the times of 0.8 s, 0.6 s, and 0.6 s, respectively. When the preset speed was 200 mm, subjects A, B, and C reached the preset training speed at the times of 1.0 s, 0.8 s, and 0.8 s, respectively. When the human leg speed was gradually stable, the training speed error of subject A was between ±2 mm/s, the training speed error of subject B was between ±3 mm/s, and the training speed error of subject C was between ±4 mm/s. It can be seen from the experimental data that the magnetic powder clutch controller could smoothly complete the training purpose and control the training speed.

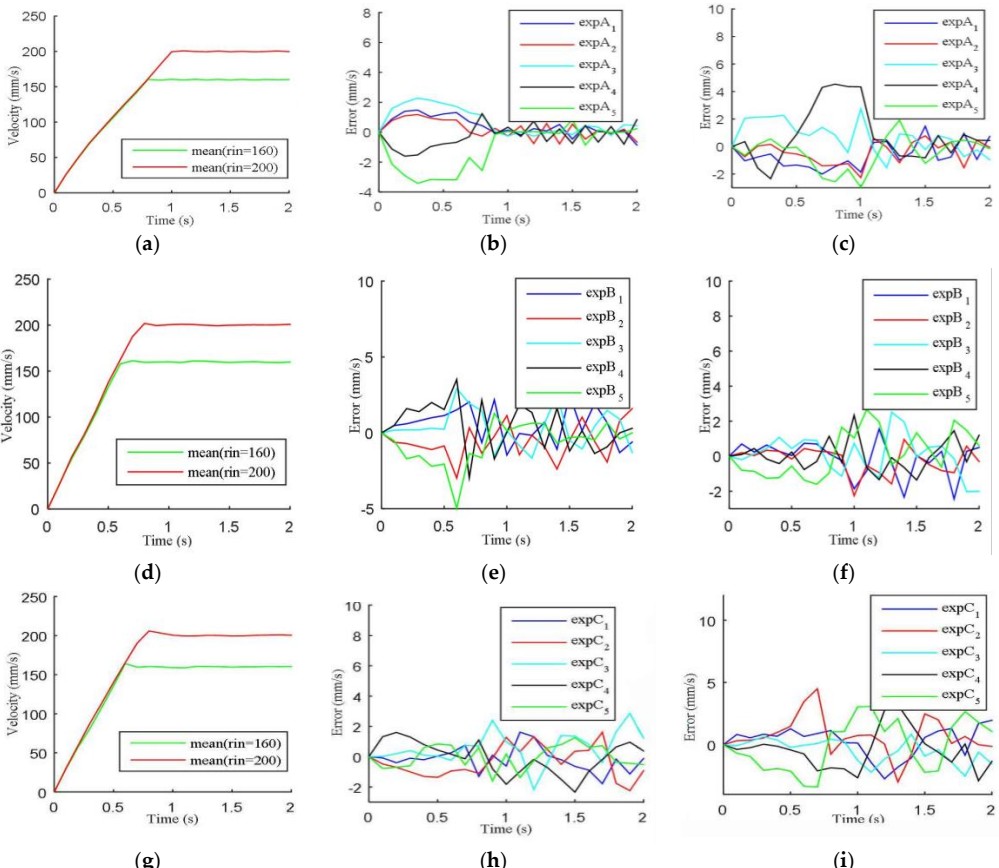

**Figure 9.** Five sets of experimental pedal speed sampling results. (**a**) Experiment A average speed difference. (**b**) Experiment A sampling speed difference (speed = 160 mm/s). (**c**) Experiment A sampling speed (speed = 200 mm/s). (**d**) Experiment B average speed difference. (**e**) Experiment B sampling speed difference (speed = 160 mm/s). (**f**) Experiment B sampling speed (speed = 200 mm/s). (**g**) Experiment C average speed difference. (**h**) Experiment C sampling speed difference (speed = 160 mm/s). (**i**) Experiment C sampling speed (speed = 200 mm/s).

## 6. Conclusions

A multi-training leg-stretching device and a new type of magnetic powder clutch with motor combination driving scheme were proposed. The dynamics model of human leg extension training was analyzed according to the principle of virtual work. The complete human dynamics model of resistance training was established, and the model equation of human dynamics was determined.

The sliding mode control algorithm with fuzzy observer anti-obstruction training based on equivalent control was designed, and the robust control was applied to the sliding mode approximation domain by means of PD control. The experiment was carried out, and the results proved the theoretical feasibility of the control algorithm.

**Author Contributions:** Methodology, H.W. and X.H.; software, Y.D. and Z.L.; validation, Y.D.; formal analysis, H.Y.; writing—Original draft preparation, H.Y.; writing—Review and editing, H.W., X.H. and Z.L.; project administration, H.W.

**Funding:** This research was funded by China Science and Technical Assistance Project for Developing Countries, grant number KY201501009; Key research and development plan of Hebei Province, China, grant number 19211820D; The forty-third regular meeting exchange programs of China Romania science and technology cooperation committee, grant number 43-2; Shanghai Science and Technology Innovation Action Plan, grant number 19441908200.

**Conflicts of Interest:** The authors declare no conflict of interest.

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
