# Peer review of "Design and Isokinetic Training Control Method of Leg Press Training Device"

_applsci, doi:10.3390/app9142822_

Reviewer 1 Report

Review comments to applsci-504515

In this research, the authors propose a leg training device with multiple modes. The control algorithm is designed based on human-device dynamic model, and validated by the constant velocity training experiments. This study might have its merits, but the manuscript has not been written in standard English and reads mystifying at intervals. Many errors occur, while important information is sometimes missing. I recommend a major revision of the manuscript, and reconfirm the validity of the study after major revision.

Major comments:

1)    The manuscript needs thorough English proofreading. I could not enumerate all those grammatical problems, but even the very first sentence in the Abstract (Line 11-14) is already problematic. 

2)    Some symbols in the equations and captions seems faulty, for examples:

Line89: J_ci;

Line90: w_p

Line84: eta_1;

Line94: a limit symbol in the upper equation of Eq.3;

Fig. 17: Captions in panel (a), (c),(d), (f), (g) and (i) are faulty. Also, is "speed difference" equal to “speed error”?

3)    Some important information in the equations seems missing, which makes those equations unintelligible, for examples:

Table 2:  What are "serial numbers"? Please explain and mark them in Fig.5; Also, what does symbol "S" mean?

Table 3: What does "f_1" and "f_2" mean?

Table 4 and Eq. 4, what does "m_h" mean? Should "m_p" in Fig. 5 be "m_h"?

Eq.6: What do "K","r_k", "sigma" and "x_m0" mean?

Eq. 7: What does the constant "125" refer to?

Eq. 24: What does "i" mean?

4)    There are two Table 2, two Table 3 and two Table 4 in the manuscript.

5)    In the second Table 2 in Line 175, the pedal acceleration of phase B should be zero; the state of consciousness of phase C should be "Stretching leg" while the state of consciousness of phase F should be "Curved leg".

6)    Please explain "certain parameters" and "uncertain parameters" in Line 182-183.

7)    Chapter 6 Discussion is such short and simple, I would suggest combining it with Chapter 5.

Minor comments:

8)    In Table 1:"Training Mode" should be set as the most left column.

9)    In Figure 3: Capitalization is applied disorderly.

10) L114: please explain "particle system".

11)  L243: What is "upper resistance"?

12)  Table 3 in Line 241: Physical meanings of "P/Z/N" and "B/M/S/O" should be explained.

13)  Table 4 in Line 248: "experimenter" should be replaced by "subject"

14)  L268-269, please add unit to the numbers.

15)  Author Contributions: please use initials rather than full names.

Author Response

The precious questions you have raised have been revised. Details will be found in the uploaded word and the latest paper.

Reviewer 2 Report

Please check again in the Latex description on ","

page 5, line 122; page 7, line166.

It is difficult to find the ethical treatment for the human subjects.

It is unclear whether 3 subjects with 5 trials are enough for the validation.

I recommend to add the average and error bar (such as standard deviation) in Figure 17.

Author Response

The precious questions you have raised have been revised. Details will be found in the uploaded word and the latest paper.

Round  2

Reviewer 1 Report

I have to say, extensive editing of the English language and style in the manuscript is still needed.

There is still a typo in Eq.3.   (limx->0)

Author Response

Response to Reviewer 1 Comments

Point 1: I have to say, extensive editing of the English language and style in the manuscript is still needed.

Response 1:  

The language of the article has been polished.

Point 2:There is still a typo in Eq.3. (limx->0)

Response 2:  

Already modified.

Thank you for your help.
